# Cationic Polystyrene-Based Hydrogels as Efficient Adsorbents to Remove Methyl Orange and Fluorescein Dye Pollutants from Industrial Wastewater

**DOI:** 10.3390/ijms24032948

**Published:** 2023-02-02

**Authors:** Silvana Alfei, Federica Grasso, Valentina Orlandi, Eleonora Russo, Raffaella Boggia, Guendalina Zuccari

**Affiliations:** Department of Pharmacy, University of Genoa, Viale Cembrano, 16148 Genoa, Italy

**Keywords:** water pollution, anionic dyes contaminants, azo dyes, electrostatic absorption, cationic styrene-based resins, high swelling capacity, rheological properties, absorption experiments, dye removal by contact, dye removal by filtration, excellent removal efficiency

## Abstract

Water pollution from dyes is harmful to the environment, plants, animals, and humans and is one of the most widespread problems afflicting people throughout the world. Adsorption is a widely used method to remove contaminants derived from the textile industry, food colorants, printing, and cosmetic manufacturing from water. Here, aiming to develop new low-cost and up-scalable adsorbent materials for anionic dye remediation and water decontamination by electrostatic interactions, two cationic resins (R1 and R2) were prepared. In particular, they were obtained by copolymerizing 4-ammonium methyl and ethyl styrene monomers (M1 and M2) with dimethylacrylamide (DMAA), using N-(2-acryloylamino-ethyl)-acrylamide (AAEA) as cross-linker. Once characterized by several analytical techniques, upon their dispersion in an excess of water, R1 and R2 provided the R1- and R2-based hydrogels (namely R1HG and R2HG) with equilibrium degrees of swelling (EDS) of 900% and 1000% and equilibrium water contents (EWC) of 90 and 91%, respectively. By applying Cross’ rheology equation to the data of R1HG and R2HG’s viscosity vs. shear rate, it was established that both hydrogels are shear thinning fluids with pseudoplastic/Bingham plastic behavior depending on share rate. The equivalents of -NH_3_^+^ groups, essential for the electrostatic-based absorbent activity, were estimated by the method of Gaur and Gupta on R1 and R2 and by potentiometric titrations on R1HG and R2HG. In absorption experiments in bulk, R1HG and R2HG showed high removal efficiency (97–100%) towards methyl orange (MO) azo dye, fluorescein (F), and their mixture (MOF). Using F or MO solutions (pH = 7.5, room temperature), the maximum absorption was 47.8 mg/g in 90′ (F) and 47.7 mg/g in 120′ (MO) for R1, while that of R2 was 49.0 mg/g in 20′ (F) and 48.5 mg/g in 30′ (MO). Additionally, R1HG and R2HG-based columns, mimicking decontamination systems by filtration, were capable of removing MO, F, and MOF from water with a 100% removal efficiency, in different conditions of use. R1HG and R2HG represent low-cost and up-scalable column packing materials that are promising for application in industrial wastewater treatment.

## 1. Introduction

Badly treated and/or untreated waste released into bodies of water causes water pollutions, which is one of the most widespread problems afflicting people throughout the world. Polluted waters are detrimental for plants and organisms living in or around the aquatic ecosystem and can also damage people and animals who consume them [1]. Water pollutants mainly consist of pathogens, inorganic compounds, organic material, and macroscopic pollutants. Among organic pollutants, dyes and pigments are complex molecules extensively used in different industries [2]. In particular, the widespread modern textile production associated with inadequate wastewater treatment has dramatically increased the discharge of toxic, carcinogenic, and alarming pollutants, such as dyes and pigments, into the environment [2]. Indeed, textile manufacturers produce 50–100 L of wastewater per kg of finished product [3]. Troublingly, it was expected that 10–15% of all dyes used in textile processes and other industries (700,000 tons per year) is discharged into wastewater, causing massive aquatic pollution [4]. Pigments are water-insoluble substances, while dyes are soluble or partially soluble in water, thus resulting in the ability to interact with the leather or fiber, imparting strong color. Their color is so intense that even very low concentrations of dyes in effluents are highly visible. When improperly cast off into aquatic ecosystems, the effluents that comprise these substances can cause serious disorders, such as the reduction of the solubility of oxygen in the water and several adverse effects due to their toxicity, mutagenicity, and carcinogenicity and/or those of their intermediates [5,6,7]. Additionally, dyes, impeding light penetration, significantly affect the photosynthetic activity in aquatic life and are also toxic to the marine and freshwater ecosystem [8], thus limiting their essential functions for life. Indeed, among their several pivotal operations, the components of the aquatic ecosystem filter dilute and store water, prevent floods, maintain climate balance locally and globally, and protect biodiversity. Furthermore, organic coloring materials, when in water, stimulate the growth of many aquatic micro-organisms that are lethal to humans and aquatic life [8]. Based on their general structure, dyes can be classified as anionic, non-ionic, and cationic substances. Dyes consist of both natural compounds and synthetic molecules, which are in turn classified into non-azo dyes and azo dyes. Azo dyes are the largest class of colorants and represent over 60–70% of dyes globally produced [9] and 50% of all the dyes used in industries [10]. Azo dyes are typified by the presence of a double bond between two nitrogen atoms (–N=N–). In azo dyes, one or both nitrogen atoms are attached to a naphthalene or benzene ring, and the presence of additional carboxyl, hydroxyl, amino, or sulfonyl functional groups confers them with amphoteric properties [11].

Azo dyes are resistant to many types of treatments and are difficult to be converted wholly or partly into a mineral or inorganic material, so, due to their bio-recalcitrance, they persist in aquatic environments. The methods that have been employed for the removal of color and toxicity from effluents polluted by dyes and azo dyes produced by textiles, food colorants, printing, and cosmetic manufacturing, include physical, chemical, and biological treatments. Coagulation–flocculation [12], advanced oxidation processes [13], biodegradation [7], and adsorption/biosorption [14] are among the most common techniques adopted. Over the last decade, nonstop research has allowed the improvement of traditional physicochemical methods to advanced oxidation processes, while innovative biocatalysts and integrated biochemical approaches, as well as biological treatments using microorganisms, have also been employed for dye remediation.

As reported, organic dyes can be removed by chemical reduction using hydrogen catalytically generated using hybrid microgels. In particular, metal nanoparticles (NPs) associated with cross-linked polymeric microgels have received great attention during the last few years for the purification of water. Recently, microgels loaded with cobalt (Co) NPs have been explored for the catalytic production of hydrogen and the reduction of nitroarenes and organic dyes [15].

Additionally, Arif et al. reported the removal by a reduction in the aqueous medium of water pollutants, including organic dyes, such as methylene blue (MB) and methyl orange (MO), using spherical core-shell microgel particles [16,17].

However, the use of adsorbents, bio-sorbents, nanofiltration, coagulation-flocculation, and ion-exchange methods has gained much recognition due to their workable operational measures, low cost, flexibility, and harmless end-products [18].

In fact, in contrast to the biodegradation processes that occur with the formation of metabolites that are, paradoxically, often more toxic than the dye to remove, the biosorption/adsorption processes occur without the formation of byproducts. Regarding this, natural materials, such as chitin, chitosan, peat, yeast, and fungi biomass, are employed as sorbents, acting as chelating and/or complexing materials to remove dyes from a solution, and their efficiency and selectivity of adsorption is mainly due to ion exchange mechanisms [19].

Ahadi et al. reported on the use of montmorillonite clay (MC) magnetically modified by starch and cobalt-ferrite (CoFe_2_O_4_) (MC/starch/CoFe_2_O_4_), to remove methyl violet (MV) and MB dyes from water by sorption [20].

Additionally, clinoptilolite/Fe_3_O_4_ (Clin/Fe_3_O_4_) nanocomposite powders and alginate/clinoptilolite/Fe_3_O_4_ (Alg/Clin/Fe_3_O_4_) nanocomposite particles were used for eliminating MB cationic dye from aqueous solution by sorption [21].

The current research is relentlessly focused on improving the specificity, robustness, and scalability of individual practices, as well as on designing novel approaches based on integrated and emerging techniques.

Recently, the efficiency of a new nanocomposite based on Cloisite 30B clay modified with ZnO and Ag_2_O NPs (Cloisite 30B/ZnO/Ag_2_O) in simultaneously eliminating CV and MB from water by degradation through a sono-photocatalytic process was reported [22].

The use of ultrafiltration (UF) is more advantageous than other types of filtrations due to reduced energy costs [23,24]. Additionally, water-soluble surfactants (WSSs) or water-soluble polymers (WSPs) can be added to further improve the retention of dyes during UF [25]. When UF is associated with WSSs, we have micellar-enhanced ultrafiltration (MEUF), while when UF is combined with WSPs, we have polymer-enhanced ultrafiltration (PEUF). Some studies have reported both the use of PEUF to remove MB dye from water and the ability of this dye to interact with different WSPs [26]. In particular, Moreno-Villoslada et al. studied, by UV–vis techniques, the type of interaction of MB with different soluble ionic polymers having in their structure the sulfonate group (SO_3_^−^), including poly (sodium 4-styrene sulfonate) (PSS), poly (sodium vinyl sulfonate) (PVS), and poly [sodium 2-(N-acrylamide)-2-methylpropanesulfonate] (PAMPS) [27]. Oyarce et al. evaluated the removal of MB using a sodium alginate (SA) biopolymer. Specifically, they investigated the effectiveness of varying the pH, SA dose, MB concentration, and SA reuse. A removal efficiency of 98% of the dye at the starting concentration of 50 mg/L was assessed at pH = 8 using 0.025 g of SA [28]. Additionally, the absorbent efficiency of several ammonium salts, in the form of water-insoluble polymers and copolymers (resins), has long been studied [29,30].

In this regard, water insoluble resins with functional groups capable of capturing metal ions and dyes from wastewater can also supply hydrogels. Hydrogels are characterized by well-defined three-dimensional (3D) porous structures and high hydrophilicity, due to the presence of hydrophilic functional groups such as carboxylic acid, amine, hydroxyl, and sulfonic acid groups. As adsorbents, hydrogels are an important tool to remove dyes and heavy metals from water because of their flexible network, which allows solutes to quickly penetrate water and form stable complexes with functional groups [31].

Additionally, hydrogels have a high swelling capability in water, thus increasing their original volume up to several times. This phenomenon leads to the exposure of their functional groups, which become more accessible, thus promoting the interaction with dyes and metal ions and their sorption onto polymer chains [32,33]. In this field, CMC-g-Poly (MAA-co-Aam)/Cloisite 30B (Hyd/C30B) and poly (methacrylic acid-co-acrylamide)/Cloisite 30B nanocomposite (poly (MAA-co-Aam)/Cl30B) hydrogels demonstrated high efficiency in adsorbing MB dye from wastewater samples [34,35].

On these considerations, the scope of this study was to develop new low-cost adsorbent materials obtainable by simple and scalable up procedure, for anionic dyes remediation and water decontamination, by simple electrostatic mechanisms. To this end, 4-ammonium methyl styrene (4-AMSTY) (here named M1) and 4-ammonium ethyl styrene (4-AESTY) (here named M2) recently reported by us [36] were copolymerized with dimethylacrylamide (DMAA) by a one-step reverse-phase suspension copolymerization technique, using N-(2-acryloylamino-ethyl)-acrylamide (AAEA) as a cross-linker. The cationic resin R1, recently described as possessing excellent gelling properties [37], and the new resin R2, were obtained and characterized by various analytical techniques, which established their structure and morphology. From the results, R2 and R1 revealed a spherical morphology, micro-dimensioned particles, and high hydrophilicity. Upon their dispersion in an excess of water, R1- and R2-based hydrogels (R1HG and R2HG) were achieved without using any other additive or gelling agent, which could be released and paradoxically contaminate water during the use of R1HG or R2HG as decontaminant adsorbents. R1HG and R2HG demonstrated high swelling capability and porosity. The rheological properties of R1HG and R2HG were assessed by measuring their viscosity vs. shear rate, which evidenced a shear thinning behavior. Additionally, by using the Cross rheological equation and a graphical and mathematical hybrid method proposed in the literature [38], the parameters of the Cross equation were found, and their goodness was verified. The curves of shear stress vs. shear rate were obtained, which decreed the pseudoplastic/Bingham plastic behavior of both R1HG and R2HG, depending on the shear rate values. The equivalents of -NH_3_^+^ groups, essential for an efficient absorbent activity by electrostatic interactions, were estimated by the method of Gaur and Gupta [39] on R1 and R2, while those of R1HG and R2HG were determined by potentiometric titrations. In absorption experiments carried out in different conditions and monitored with UV-Vis methods, R1HG and R2HG showed high adsorption efficiency based on electrostatic interactions, both by contact (97–100%) and by filtration (100%) towards the methyl orange (MO) azo dye and fluorescein sodium salt (F) selected as models of anionic dyes, as well as towards the mixture of MO and F (MOF).

## 2. Results and Discussion

### 2.1. Synthesis of R1 and R2

The cationic monomers M1 and M2 (Figure 1) were synthesized as previously reported [37].

The characterizations data obtained by using analytical techniques such as those reported in Alfei et al. [36] confirmed their structure. Then, according to the low-cost, one-step procedure known as reverse suspension co-polymerization, which was recently optimized by us to convert M1 in the resin R1 [37], both M1 and M2 were transformed in the corresponding resins R1 and R2 (Figure 1).

In particular, M1 and M2, co-monomer DMAA, and the cross-linker AAEA, previously dissolved in water, were suspended in a mixture of CCl_4_/hexane. SPAN 85 and ammonium persulfate (APS)/tetramethylethylenediamine (TMEDA) were added as anti-coagulant and initiators, respectively.

The conditions of polymerizations of M2 were here optimized as previously done for M1 [37], and the data reported in the Section 3 are those of two representative reactions. Reactors with a cylindrical geometry, which are able to minimize the horizontal component of the stirring motion of the suspension, were used to reduce the tendency of the micro-drops to aggregate [37]. After 90 min, the polymerization was interrupted, and the precipitate R1 and R2 were separated by filtration. R1 and R2 were washed sequentially several times with isopropanol, chloroform, water, ethanol, and acetone. R1 and R2 were brought to constant weight at vacuum and stored at room temperature (r.t.) for the subsequent operations of sieving and characterization. The conversion yields were >98%.

### 2.2. Sieving of R1 and R2 and Optical Microscopy

R1 and R2 were fractioned by sieving using sieves of 35–120 mesh, obtaining beads with mass equal to the 98.3% (R1) and to the 96.2% (R2) of the original weight. The microstructure of these particles was investigated by optical microscopy and scanning electron microscopy (SEM). Optic micrographs, such as those reported in Appendix A and in the following Figure 2a,b, were obtained for R1 and R2, respectively. An SEM image of R1 is available in Appendix A, while two SEM images of R2 are observable in Figure 2c,d.

As previously reported [37], the particles of R1 appeared as micro-spherical beads with size in the range 125–250 µm (Appendix A). Similarly, the two representative optical microphotographs obtained with resin R2 evidenced particles of 171 µm (Figure 2a) and 166 µm (Figure 2b). Similar to particles of R1 (Appendix A), those of R2 (Figure 2a,d) appeared both in optic images and in those from SEM significantly polydisperse.

### 2.3. Estimation of the Equivalents of -NH_3_^+^ Contained in R1 and R2

The equivalents of -NH_3_^+^ contained in resins R1 and R2 were estimated by the method of Gaur and Gupta, which was recently used and described by us to analyze R1 [37,39]. According to the literature, this method is specific for determining the NH_2_ group’s content in insoluble matrices and is operator-friendly and sensitive [37,39]. In particular, the amino groups were marked with residues of 4-O-(4,40-dimethoxytriphenylmethyl)-butyryl, and the quantitative determination of the 4,40-dimethoxytriphenylmethyl cation (ε = 70,000 at 498 nm) released from the resin after treatment with HClO_4_ was carried out by UV-Vis spectroscopy [39]. The NH_2_ moles for the grams of resins were estimated from the values of absorbance (A) determined at 498 nm, as described in the Section 3.

According to the results, expressed as the mean of three independent determinations ±standard deviation (SD), the NH_2_ equivalents present in R1 were slightly higher than those previously determined [37], being 13.20 ± 0.062 mmol/g, while those present in R2 were 16.15 ± 0.062 mmol/g.

### 2.4. Preparation of R1HG and R2HG

Recently, we reported the potent antibacterial/bactericidal effects of a polystyrene-based cationic resin (R4), structurally similar to R1 and R2. Particularly, R4 demonstrated to reduce efficiently (>99%) the bacterial contamination in water both by contact and by filtration, by killing pathogens upon electrostatic interactions with their negatively charged surface [40]. More recently, resin R1 was demonstrated to have excellent gelling properties and was used to formulate two antibacterial pyrazoles, providing Bingham pseudoplastic hydrogel formulations promising for topical administration [37]. Since it was not in the scope of our recent publication on pyrazole formulation, the empty gel R1HG obtained in preliminary investigations on R1 was not characterized. Here, aiming to develop cationic 3D materials to remove anionic dyes from industrial wastewater by electrostatic interactions, R1 and R2 were successfully converted into two cationic single-component hydrogels (R1HG and R2HG). We were inspired by the proved capability of the cationic polystyrene resin R4 to bind anionic surfaces such as those of bacteria and relied on the capabilities of R1 to self-form stable hydrogels in water. No additional additives, such as stabilizers or gelling agents, were employed, thus avoiding the possibility that, paradoxically, during the use of our materials to decontaminate water, extra contaminants could be released. According to the procedure described in the Section 3, we prepared R1HG and R2HG containing the maximum amount of water that R1 and R2 were capable of absorbing. Table 1 contains the experimental data of the preparation of R1HG and R2HG from resins R1 and R2, their EWC and EDS, as well as details about R1 and R2 concentrations in the prepared hydrogels.

Appendix A shows the obtained very viscous and semi-transparent hydrogels, R1HG (left side) and R2HG (right side), in inverted positions and on watch glasses, whose volume and weight corresponded to those of gels at their EDS of 900 and 1000%. Figure 3 shows representative microphotographs of the swollen R1HG (a, b) and R2HG (c, d).

As reported in Table 1, the EWC of R1HG and R2HG (≥90%), as well as their EDS values [900% (R1) and 1000% (R2)], were similar and very high. In Figure 3, it can be observed that large particles of the swollen R1HG and R2HG appeared to be unequivocally filled with water and more than 2-fold larger than the particles of R1 (Appendix A) and R2 (Figure 2).

### 2.5. Attenuated-Total-Reflectance (ATR) Fourier Transformed Infrared (FTIR) Spectra

The ATR-FTIR analyses were carried out on the original resins (R1 and R2), on the gels prepared dispersing R1 and R2 in water (R1HG and R2HG), and on the fully dried gels (D-R1HG and D-R2HG) obtained by gently heating freshly prepared R1HG and R2HG for about 7 h. The analysis was carried out in triplicate for each sample, and representative images of all the obtained spectra are shown in Figure 4a (R1, R1HG, and D-R1HG, as in the legend) and Figure 4b (R2, R2HG, and D-R2HG, as in the legend).

As expected, the ATR-FTIR spectra of R1 and R2 were very similar and showed the weak CH stretching band of methylene groups (2930 and 2927 cm^−1^ respectively) and the weak and large NH stretching band (3399 and 3400 cm^−1^ respectively) due to the M1 and M2 contribute. Additionally, strong bands at 1609 and 1612 cm^−1^, typical of di-methyl-acrylamide derivatives, such as DMAA and AAEA, were detected, thus assessing the presence of all three main ingredients in the structure of R1 and R2. No band was observable in the range of 900–911 cm^−1^, (H_2_C=CH- vinyl group), thus confirming the absence of residual monomers. The ATR-FTIR spectra of the soaked R1HG and R2HG were very simple, showing only the typical bands of water, including large OH stretching bands over 3000 cm^−1^ and OH scissoring bands at 1636 cm^−1^, thus confirming the very high content of water (>92%) in both R1HG and R2HG. On the contrary, the spectra of the dried samples obtained by heating the soaked gels were very similar to those of R1 and R2, thus establishing that the interactions that occurred during the formation of the gels did not affect the main functional groups of original resins and were reversible. In particular, weak bands at 1496–1498, 1401–1402, 1254, and 1141 cm^−1^ provided by the aromatic C=C and C-N stretching bands, and intense bands at 1612–1614 cm^−1^ standing for the C=O stretching of the amide group of DMAA and AAEA were detectable in both spectra.

#### Principal Components Analysis (PCA) of ATR-FTIR Data

First, a matrix of 20,406 variables was constructed collecting the spectral data (wavenumbers) of R1, R2, R1HG, R2HG, D-R1HG, and D-R2HG. Secondly, the resulting dataset was first pre-treated by standard normal variate (SNV) normalization and then processed using the PCA. PCA is a potent chemometric tool capable of extracting essential information from a complex and vast set of correlated variables by reducing them to a limited number of uncorrelated variables, namely Principal Components (PCs) [37,41,42,43,44,45]. PCA can provide results in the form of loadings plots, score plots, and biplots.

Particularly in the case of score plots, PCA allows us to visualize the reciprocal positions occupied by the analyzed samples in the orthogonal space of two selected PCs, where the scores are the new coordinates of samples in the PCs’ space. Specifically, the score plot shows the behavior of samples in such space, highlighting the similarities and differences in their chemical composition. Here, the score plot of the six analyzed samples (PC1 vs. PC2) is shown in Figure 5.

As can be observed, although PC 2 could also be seen a sort of separation according to the minimal structural differences existing between the compounds of family n = 1 (located at positive scores) and those of family n = 2 (located at scored close to zero or even highly negative), the samples were well separated on PC1 on the base of their water content. In fact, while original resins R1 and R2 and the dried gels were all clustered at positive scores, the soaked gels were grouped at negative scores.

### 2.6. Weight Loss (Water Loss)

Appendix A shows the appearance of the fully dried D-R1HG (right side) and D-R2HG (left side) obtained by heating R1HG and R2HG at 37 °C for 465 min. As can be observed from Appendix A, when R1HG and R2HG were heat-dried, they provided both porous amorphous solids. Figure 6a shows the curves obtained, reporting in a graph the values of the cumulative weight loss (%) of R1HG and R2HG vs. times. Figure 6b,c shows the kinetic models that best fit the data of cumulative weight loss curves in Figure 6a.

As observable in Figure 6a, the weight loss was almost quantitative (94%) for both R1HG and R2HG, and the equilibrium was reached after either 6 h 45 min (R1) or 5 h 45 min (R2). To know exactly the kinetics and the main mechanisms that governed the loss of water from the R1HG and R2HG, we fit the data of the curves in Figure 6a with the zero order model (% cumulative water release vs. time), first-order model (Log_10_ % cumulative water residuals vs. time), Hixson–Crowell model (cube root of % cumulative water residuals vs. time), Higuchi model (% cumulative water release vs. square root of time), and Korsmeyer–Peppas model (Ln% cumulative water release vs. Ln of time) [37,46,47,48,49]. The kinetic models were so obtained as dispersion graphs, and the coefficients of determination (R^2^) of the linear regressions of the obtained dispersion graphs were the parameters to determine which model best fit the water release data. R^2^ values are reported in Table 2 and proved that the water loss from R1HG best fit with the first-order kinetic model (Figure 6b), while that from R2HG fit with the Korsmeyer-Peppas one (Figure 6c), with the R2 values being the highest ones.

First-order kinetics states that the change in concentration of a released substance with respect to the change over time is dependent only on the residual concentration of the compound released. In the present case, the release of water (weight loss) over time depended only on the residual concentration of water after the heating periods [46]. First-order kinetics are described by the Equation (1):(1)$$LogQt=\left(\frac{K}{2.303}\right)\times x+LogQo$$
where *Qt* is the amount of water released on time *t*, *Qo* is the initial amount (%) of water in the gel, and *K* is the first-order constant. Accordingly, *K*/2.303 corresponded to the slope of the equation of the linear regression of our first-order mathematical model shown in Figure 6b, while *Log Qo* was its intercept. Consequently, the first-order constant was negative and equal to −0.0069 and the original percentage content of water in the weighted R1HG was of 107%. As for the Korsmeyer-Peppas kinetic model, it is given by the Equation (2):(2)$$\frac{Mt}{M\infty }=Ktn$$
where *Mt* is the amount of water lost at time *t*, *M∞* is the total amount of water released at infinite time, *K* is the release rate constant, and *n* is the release exponent, indicating the type of release mechanism. In the equation of the linear regression in Figure 6c, *n* corresponded to the slope, Ln *M∞* was the intercept [50]. In our case, *n* was 0.7432 and in the range 0.5–1, thus indicating a combination of different mechanisms and establishing that the loss of water by R2HG system was governed by a non-Fickian anomalous diffusion [50].

### 2.7. Equilibrium Swelling Rate

The swelling measurements were made at fixed times following the procedure described in the Section 3 and according to indication of literature [51]. Figure 7 shows the cumulative swelling ratio percentage curves of R1HG and R2HG.

As can be observed, the equilibrium swelling ratio (Q_equil._), which was determined at the point the hydrated resins achieved a constant weight, was reached after only 10 min by R1 and 20 min by R2, establishing for values of Q_equil._ = 1226 (R1HG) and 1669 (R2HG).

### 2.8. Potentiometric Titrations of R1HG and R2HG

Following potentiometric titrations carried out as described in Section 3, by graphing the measured pH values vs. the aliquots of HCl 0.1N added, the titration curves were obtained (Figure 8, lines with round indicators). Successively, the dpH/dV values were computed from titration data and were reported in the same graph vs. those of the corresponding volumes of HCl 0.1N. The first derivative lines (1stD) of the titration curves (Figure 8, lines with square indicators), whose maxima represent the titration end points, were achieved. The potentiometric titrations of R1HG and R2HG allowed both to assess their titration profile, thus detecting how many types of basic centres exist in gels and titrating the NH_2_ groups contained in R1HG and R2HG to determine their NH_2_ content per gram of gel.

As observable in Figure 8, while R1HG showed a significant buffer capacity of up to 3 mL HCl 0.1 N added and one end titration point upon the addition of 3.5 mL HCl 0.1N (maximum 6.12 in the first derivative curve), R2HG showed two jumps in the titrations curve. In particular, the first was smaller and corresponded to a small maximum (2.74) in the first derivative curve at the addition of 2 mL of HCl 0.1N, while the second was more significant, corresponded to the highest maximum (5.98) in the first derivative curve, and occurred upon the addition of 3.5 mL HCl as in the case of R1HG. Table 3 contains the results related to the determination of the equivalents of the NH_2_ groups contained in R1HG and R2HG.

As previously evidenced by the UV-Vis determinations on dry resins using the method of Gaur and Gupta, the results from potentiometric titrations also confirmed that the equivalents of NH_2_ are higher in the material prepared using the monomer M2 (n = 2) than in that prepared making use of monomer M1 (n = 1). However, according to Table 3, the NH_2_ groups contained in 1 g of hydrogels were 0.9972 mmol/g in R1HG and 1.0373 mmol/g in R2HG. Considering that, according to the data reported in Table 1, the content of cationic resins R1 and R2 in 1g of the correspondent gels is 0.076 g (R1) and 0.064 g (R2), the results obtained by the potentiometric titrations of gels fit perfectly with those obtained by the method of Gaur and Gupta analyzing the dried resins R1 and R2. In fact, an NH_2_ content of 0.9972 and of 1.073 mmoles per gram of gels should correspond to 13.12 and 16.21 mmoles per gram of resin, respectively, against the determined value of 13.20 and 16.15 with errors of 0.6% and 0.4%.

### 2.9. Rheological Studies

The physicochemical characterization of R1HG and R2HG was completed performing rheological experiments. The values of their apparent viscosity (η [Pa × s]) as a function of the applied shear rate (γ [s^−1^]) were first determined, and by plotting η values vs. γ values, the curves shown in Figure 9a (R1HG) and Figure 9b (R2HG) were achieved.

As expected, the viscosity of R2HG was significantly lower than that of R1HG, due to the lower concentration of R2 and the higher content of water of R2 HG. However, for both gels, it was seen that η decreased considerably for small increases of γ up to values of γ < 30, while η was practically constant and did not change significantly for values of γ > 30. Both gels were demonstrated to be non-Newtonian fluids not following Newton’s law of viscosity. The graphical representation of Newton’s law expressing the linear relationship between shear stress (τ) and γ, is a line with slope η and intercept zero, where η is constant and not dependent on γ [52].

On the contrary, in non-Newtonian fluids, viscosity is γ-dependent. Generally, non-Newtonian fluids that display a decreased viscosity with increasing γ (as in the present case) are defined as shear-thinning fluids, having values of the flow behavior index *n* < 1. Differently, those fluids that for high values of γ exhibit a viscosity increasing with increasing γ are termed shear thickening dilatant fluids, having *n* > 1. For values of γ up to 100, R1HG and R1HG behaved as shear thinning fluids.

Additionally, non-Newtonian shear thinning fluids may be Bingham plastic fluids, pseudoplastic fluids and Bingham pseudoplastic fluids. In the Bingham plastic fluids viscosity is constant, the relationship between τ and γ is linear, but the intercept is >0. In pseudoplastic fluids and Bingham pseudoplastic fluids, viscosity is not constant and their graphs of τ vs. γ are not linear and have either intercept 0 (pseudoplastic fluids) or >0 (pseudoplastic Bingham fluids). In particular, pseudoplastic and pseudoplastic Bingham fluids over certain values of shear rate could behave as plastic Bingham fluids with η independent from γ. Here, to exactly assess the rheological behavior of R1HG and R2HG, we determined the values of n and studied the relationship between γ and τ. To this end, we initially used Cross’ rheology equation expressed in the forms reported in the study by Xie and Jin [38]. In particular, we considered first Cross’ equation (3), which includes four parameters necessary to predict the general flow behaviors of non-Newtonian fluids [38].
(3)$$\eta =\frac{\eta o+\eta \infty \times \alpha \gamma^{n}\;}{\left(1+\alpha \gamma^{n}\right)}$$

In Equation (3), *γ* is the shear rate, *η*o is the viscosity when the shear rate is close to zero, *η∞* is the viscosity when the shear rate is infinity, *n* is the flow behavior index, and *α* is the consistency index.

Following the hybrid method reported in the literature [38], we determined the four parameters of the Cross equation (3) and checked the accuracy of the parameters estimates by fitting them to our experimental data. Initially, the flow behavior index *n* was determined by a graphical method using the following Equation (4):(4)$$Log\;\eta =Log\;\left(\frac{\eta o}{\alpha }\right)-nLog\gamma$$
and plotting our measured data as *Log η* vs. *Log γ* [Appendix A].

Since *Log η* is linear with *Log γ*, *n* can be obtained as the slope of the linear relationship of *Log η* and *Log γ*.

The equations of the linear regressions obtained for both R1HG and R2HG and the related coefficients of determination (R^2^) have been reported in Table 4.

The values of *n* confirmed the shear thinning behavior of our gels being <1. The viscosity at the very high shear rate (*η∞*) was determined by another graphical method converting Equation (3) into the form of the Equation (5) below.
(5)$$\eta =\eta \infty +\frac{\eta o\;\;}{\alpha \;\;}\gamma^{-n}$$

It can be observed that the relationship between η and *γ^−n^* is linear, and *η*∞ is the intercept of the equations of the linear regression line obtained by plotting the values of η vs. those of *γ^−n^* [Appendix A].

The equations of the linear regressions obtained, and the related correlation coefficients (R^2^) have been reported in Table 4. The viscosity at a very low shear rate (η0) was determined by the relationship of ηo = 1000 η*∞*, as reported in the literature [38].

Finally, the consistency index *α* can be obtained either by the intercept values of the linear equations reported in Appendix A or by the slopes of the linear equations reported in Appendix A. The values of *n*, *ƞ∞*, *ƞ*o, and *α* have been included in Table 4. Since an ideal rheology equation relating viscosity and shear rate should provide an accurate fit for most experimental measurements over a wide range of the shear rate change [38], we verified the goodness of the estimated parameters by determining the viscosity values for R1HG and R2HG according to the Cross equation and the estimated parameters reported in Table 4. From Figure 10a,b, it can be seen that the viscosity and shear rate relationships obtained (yellow lines) are in reasonable agreement with the experimental results (round indicators).

Then, using the following form of Cross equation [53] [Equation (6)], which contains three parameters (consistency *α*, exponent *n*, and reference shear stress *τ*), and Equation (3), the shear stress values as a function of shear rate were obtained, resolving the Equation (7), and the plots of shear stress vs. shear rate were reported in Figure 11.
(6)$$\eta =\frac{\alpha }{\left(1+{\left(\frac{\alpha \gamma }{\tau }\right)}^{1-n}\right)}$$
(7)$$\left(1+{\left(\frac{\alpha \gamma }{\tau }\right)}^{1-n}\right)=\left(1+\alpha \gamma^{n}\right)$$

According to Figure 11, both of the gels were pseudoplastics fluids without constant η for shear rate values <30. Over 30, their values of η were practically constant, and they assumed a plastic Bingham behavior. By the intercept of the linear tendency lines in the linear tracts of the graphs in Figure 11, the yield stress or tangential stress values were obtained for both gels and were reported in Table 4. The yield stress or yield point of a ductile material is defined as the value of the stress at which the material begins to deform plastically, transiting from an elastic behavior allowing for reversible deformations to a plastic behavior characterized by the development of irreversible deformations.

### 2.10. Removal of Dyes from Water by R1HG and R2HG

The removal of organic dyes from wastewater is an extremely important issue for the industrial emission of sewage. Contaminant dyes, mainly azo dyes, not only deteriorate the water quality but are also harmful to human health due to their toxic, carcinogenic, mutagenic, and/or teratogenic effects [54].

Here, we tested the ability of cationic hydrogels R1HG and R2HG, which self-form when R1 and R2 are added to aqueous solutions, to absorb anionic dyes by electrostatic interactions.

To study the capability of our hydrogels to absorb and retain anionic dyes, as a model of anionic organic dyes, we first select the methyl orange azo dye (MO) for its relevance as a possible hazardous contaminant of wastewater from textile industries, and then we select fluorescein (F) as representative of dyes not belonging to the azo dye family. Azo dyes are water-soluble, yielding highly colored solutions. The use of several azo dyes, such as Congo red, has long been abandoned, primarily because of their carcinogenic properties [55]. MO, which is currently used as an indicator for acid-base titrations due to its capability to change color dependent on pH, could have mutagenic and carcinogenic effects [56]. F is used in the diagnosis of herpetic corneal infections, corneal abrasions, and corneal ulcers. The sodium salt adopted here is mainly used as a tool in the fields of optometry and ophthalmology to diagnose vascular disorders or to find many muscular ventricular septal defects during open-heart surgery [57]. Along with its desired effects, fluorescein may cause some unwanted effects, which may need medical treatment. Importantly, the use of fluorescein in patients with a history of bronchial asthma may increase the risk of adverse respiratory reactions [58].

Experiments were performed both in bulk to assess the efficiency of R1 and R2 to remove dyes by contact and in columns filled with R1HG and R2HG to assess their removal efficiency by filtration. The experiments in column were performed both with freshly prepared gels and with already used gels, while both experiments in bulk and in column were carried out using dyes singularly and in mixtures.

The concentration of dyes in the original appositely prepared aqueous models of wastewater contaminated by dyes and in the water solutions treated with R1HG and R2HG were determined by UV-Vis spectrometry using calibration curves after necessary dilutions. The calibration curves were constructed by the last square methods using Microsoft excel software and standard solutions of MO and F prepared at five different concentrations in order to have values of absorbance <1. The obtained linear regression models and the related equations are shown in Appendix A.

#### 2.10.1. In Bulk Experiments: Removal of Dyes from Water by R1HG and R2HG upon Contact

As recently reported [54], typical experiments consisted first of preparing separated aqueous solutions of MO and F (50 mg/L) and an aqueous solution containing both MO and F (MOF), prepared by diluting to 250 mL, 25 mL of MO 500 mg/L and 25 mL of F 500 mg/L. The UV-vis spectra of the opportunely diluted [1:10 (MO and F), 1:20 (MOF)] untreated solutions (time T0) were measured to assess the actual concentrations of dyes obtained. The prepared solutions were added with R1 or R2 (about 1 g/L) at r.t. and left under stirring for the time necessary for the experiment (120 min). The UV-Vis spectra of solutions under treatment were measured at different time points (T5, T10, T20, T30, T60, T90, T120), after dilutions 1:10 (MO and F) and 1:20 (MOF). Using the absorbance data and the related concentrations of MO, F and of MO + F in MOF, we calculated the removal efficiency of both resins (*R%*), the amounts of dye adsorbed per unit mass of R1 and R2 (mg/g) at the fixed times (*Qt*), and the amounts of adsorbed dye by R1 and R2 at equilibrium (*Qe*). Appendix A shows the appearance of MO solutions at time T0 (untreated solutions, vial 1 on the left) and of MO solutions during treatment with R1 (a) and R2 (b) after 5, 10, 20, 30, 60, 90 and 120 min (vials 2–8 from the left to right). As in Appendix A show the appearances of F solutions and MOF solutions, respectively, obtained by experiments performed in the conditions described above. As an example, in Appendix A the UV-Vis spectra of MOF solutions at time T0 and during treatment with resins have been reported concerning R2 (Appendix A) and R1 (Appendix A). While the solutions at time T0, before contact with resins, showed intense peaks with maxima of absorbance around 465 nm (MO) and 491 nm (F), the spectra of solutions under treatment did not show detectable peaks already after only 5 min of contact with resins. To have more precise knowledge of the efficiency of our resins in removing dyes from water by contact, the values of removal efficiency (R%) of R1 and R2 were calculated for all dye solutions (MO, F and MO + F in MOF) and were plotted vs. times, obtaining the cumulative removal percentage graphs reported in Figure 12.

As can be observed, in all experiments, both resins demonstrated very high removal efficiency (96–100% for R1; 97–100% for R2) in removing dyes from water, with R2 being remarkably more efficient and rapid than R1, as later confirmed while performing kinetic studies. The higher efficiency of R2 probably was due to the presence of an additional methylene group in the carbon linker between the phenyl ring and the -NH_3_^+^ group. Being more spaced from the ring, the cationic group is freer to electrostatically interact with the negative charges of the dyes, thus increasing the absorption efficiency of R2 compared to R1.

Additionally, R2 could remove dye more efficiently than R1 because of its higher content of cationic -NH_3_^+^ groups, which could allow stronger electrostatic interactions with anionic dyes. The dye that was most difficult to remove was the not-mixed MO, for which total removal was not reached (Figure 12a). Regardless, a removal level of 97% was obtained with R2 after 30 min, and a removal level of 96% was obtained with R1 after 120 min. Fluorescein was totally removed from water after only 5 min by R2 and after 90 min by R1 (Figure 12b). Curiously, when MO and F were present in water simultaneously in a mixture, the removal of both MO and F was easier, faster, and total for both resins. In particular, R1 absorbed 100% of MO and F after only 10 min (Figure 12c), while R2 only took 5 min (Figure 12d). Curiously, in the removal of MO and F by R2, very slow reductions in the removal efficiency were observable after the achievement of the maximum value. Performing Student’s *t*-test established that there was no significant difference between percentages obtained after reaching the maximum values and the maximum value itself. Such fluctuation could be due to the UV-Vis instrument, which, in fact, provides absorbance values affected by standard deviations, which also justifies the presence of values over 100. Using the equation reported in the Section 3, we obtained the amounts of dye adsorbed per unit mass of R1 and R2 (mg/g) at the fixed times (*Qt*), which were plotted vs. times obtaining the absorption kinetic curves reported in Appendix A, which were perfectly stackable to those of cumulative removal efficiency in Figure 12. As observable in Appendix A, the amount (mg) of the not-mixed MO removed per gram of resin was lower than that of the not-mixed F for both resins, with R2 being the more efficient absorbent. In particular, although a great part of the dyes was removed after 5–10 min of contact with resins, in the case of MO, R2 reached the equilibrium adsorption state (*Qe*) after 30 min, removing 48.5 mg/g MO, while R1, after 120 min, succeeded in removing 47.7 mg/g MO. In the case of F, R2 reached the equilibrium adsorption state after 20 min, removing 49.0 mg/g F, while R1, after 90 min, succeeded in removing 47.8 mg/g F. In the case of MOF, the amounts of F and MO absorbed per gram of R1 and R2 at the equilibrium (*Qe*) were very similar, and the amounts of F were higher than those of MO (Appendix A). Additionally, the equilibrium absorption state was reached earlier by R2 than R1. In particular, R1 reached the equilibrium absorption state for both dyes after 90 min, removing 65.3 mg/g F and 40.3 mg/g MO (Appendix A), while R2 absorbed 65.2 mg/g F after 60 min and 40.6 mg/g MO after only 30 min (Appendix A).

##### ATR-FTIR Spectra of R1 and R2 after Absorption

With the aim of assessing if the presence of the absorbed dyes could be at least qualitatively detected by ATR-FTIR spectroscopy, we acquired ATR-FTIR spectra on samples of R1 and R2 recovered after treatments of the solutions of F, MO, and MOF when *Qe* was reached.

Particularly, the resins holding dyes were filtered under vacuum using a Büchner funnel equipped with a fritted glass disc and lyophilized. Then, the dried samples were analyzed using a Spectrum Two FT-IR Spectrometer (PerkinElmer, Inc., Waltham, MA, USA) in transmission mode. For comparison, the ATR-FTIR spectra of F, MO, and of a physic mixture of F and MO (MOF) were acquired in the same conditions.

Spectra were acquired in triplicate, and representative images of the spectra of F, MO, MOF, R1/F, R1/MO, R1/MOF, R2/F, R2/MO, and R2/MOF are available in Appendix A. As expected, in region 1610–620 cm^−1^ of the spectra of resins R1 and R2 after the absorption of F, MO, or MOF, many more bands than those present in the spectra of original resins R1 and R2 (Figure 4a,b) were observable, belonging to the adsorbed dyes.

However, to obtain more reliable information on the chemical composition of resins R1 and R2 after the absorption of dyes, the matrix of 37,411 variables collecting the spectral data (wavenumbers) of R1/F, R1/MO, R1/MOF, R2/F, R2/MO, R2/MOF, F, MO, MOF, R1, and R2 was first pre-treated by SNV normalization and then processed using the PCA. The score plot of the 11 analyzed samples (PC1 vs. PC2) has been shown in Appendix A.

As observable in Appendix A, the samples were clustered in groups that appeared well separated both on PC1 and PC2, according to the chemical structure and functional groups of their components. Specifically, non-polymeric dyes F, MO, and MOF were located at negative scores (red circle), while all polymeric materials, including R1, R2, R1-F, R2-F, R1-MO, R2-MO, R1-MOF, and R2-MOF were positioned at positive scores (blue circle). Additionally, while resins containing dye were all clustered in a group at score ≤0 on PC2 (green circle), resins R1 and R2 were positioned outside such group at positive scores, due to the absence of dyes in their structure (purple circle). In fact, on PC1 materials were separated on the base of their dye content. In particular, dyes or materials containing dyes were located at scores ≤0, while materials not containing dyes were positioned at positive scores.

##### Kinetic Studies

To obtain valuable information on the pathways and mechanisms of adsorption reactions, we carried out a kinetic study [59]. During the removal of F, MO, and of F and MO in MOF using R1 and R2, kinetic models of pseudo-first order (PFO) [Equation (8)], pseudo-second order (PSO) [Equation (9)], and intra-particle diffusion (IPD) or Higuchi [Equation (10)] were studied [59]:(8)$$\ln\left(Qe-Qt\right)=lnQe-K1\times t$$
(9)$$\frac{t}{Qt}=\frac{1}{K2\times Qe^{2}}+\frac{1}{Qe}t\;$$
(10)$$Qt=Kint\times t^{0.5}+I$$
where *Qe* (mg/g) and *Qt* (mg/g) are the dye adsorption capacities of R1 and R2 at equilibrium and at time *t* respectively, *K*1 is the adsorption constant of the PFO kinetic model (1/min), *K*2 is the equilibrium constant velocity of the PSO kinetic model (g/mg×g), *Kint* is the IPD rate constant (mg/g×min^0.5^), and *I* is the intercept of the linear curve (mg/g). Values of ln *(Qe-Qt)*, *t/Qt,* and *Qt* were plotted vs. times, times again, and the root square of times, respectively. Dispersion graphs were obtained, and their linear regression lines were provided by Microsoft Excel software using the Ordinary Least Squares (OLS) method. The coefficients of determination (R^2^) of all the equations of the linear regressions obtained have been reported in Table 5 and were the parameters for determining the kinetic models that best fit the data of the absorption processes and the mechanisms that governed the absorption reactions.

The results showed that in all cases, the R^2^ values for the PSO models were higher than those obtained for both the PFO and IPD models. Consequently, the kinetic behavior of the absorption of F, MO, and of F and MO in an MOF mixture using both R1 and R2 as adsorbents followed the PSO model.

Figure 13 shows the PSO kinetic models of F removal using R1 and R2, of MO removal using R1 and R2, and of the removal of F + MO in a mixture (MOF) using R1 and R2.

In the processes governed by the PSO kinetics, the velocity limiting step is considered chemical adsorption, including adsorption through the sharing or exchange of electrons between the adsorbent and the adsorbed [60] or electrostatic interactions, which were confirmed to be the main mechanism by which both R1 and R2 absorbed anionic dyes.

The values of *Qe* (cal) and *K2* were computed using the values of the slopes and intercepts of the equations in Figure 13 for all experiments and included in Table 6.

The values of *K2* established that R2 was able to remove F and MO and F + MO faster than R1 and that the removal of F was more rapid than that of MO, both when F and MO were treated alone and when they were in a mixture.

#### 2.10.2. In Column Experiments: Removal of Dyes from Water by R1HG and R2HG upon Filtration

To mimic a decontaminant column system to absorb dyes working by filtration, a plastic syringe loaded with R1HG or R2HG was used for a miniature-scale treatment of aqueous samples artificially contaminated with MO, F, or MOF. In a typical procedure, a certain volume of R1 or R2 was inserted into the plastic syringe equipped with a small plastic bottom cap (Appendix A). The entire volume of the syringe was filled with water, and the resins were allowed to absorb water and swell at their EDS. The excess not-absorbed water was removed from the bottom of the column apparatus, first by gravity and then by applying compressed air from the top. Appendix A shows some pictures of a typical column apparatus loaded with hydrogels. By controlling the initial volume of R1 and R2, we could easily control the final volume of hydrogels and, consequently, the length of the gel-columns (Appendix A).

To test the performance of the gel columns for dye adsorption, three experiments were carried out. Initially, R1-based and R2-based gel columns loaded with 8 mL hydrogels were utilized to remove first MO and then F from water (8 mL, containing 50 mg/L MO or F), refreshing the gel in the columns before the second filtration. Secondly, new separate solutions (8 mL) of MO and F (50 mg/L each) were filtered on the column gels without refreshing them before the second filtration but collecting the filtrates separately. Then, freshly prepared gel columns were further used to remove from water a mixture of MO and F (we used solutions prepared by diluting with water 1 mL MO 500 mg/L and 1 mL F 500 mg/L in a 10 mL volumetric flask).

The filtration was finished by applying compressed air to the top of columns up to completely empty the column, observing colorless filtrate in all cases. As an example, the Appendix A contains a series of images of a typical filtration experiment, using R2 to prepare the gel column. In this specific experiment, the solutions (8 mL) of MO and F (50 mg/L each) were filtered on the column gels without refreshing them before the second filtration and before collecting the clear filtrates separately (Appendix A). All of the original dye solutions and all of the filtrates were properly diluted (1:10 MO and F, 1:20 MOF) and analyzed by UV-Vis to assess the actual concentrations of dyes obtained in the prepared dye solutions, to confirm the removal of dyes, and to determine the removal efficiency (R%). In all of the experiments, the characteristic absorption peaks of dyes disappeared in the UV-vis spectra of water solutions filtered through both R1HG and R2HG (Figure 14), while the determined concentrations of dyes and the removal efficiency percentages (100% in all cases) have been reported in Table 7.

These results demonstrated that the as-fabricated gel-columns possess a strong capability to remove anionic organic dyes from the wastewater on a miniature scale and might be a promising candidate to be applied in industrial wastewater treatment.

## 3. Materials and Methods

### 3.1. Chemicals and Instruments

Methyl orange (MO) and fluorescein sodium salt (F) were purchased by Merk Life Science S.r.l. (Milan, Italy). All other reagents and solvents were from Merk (formerly Sigma-Aldrich) and were purified by standard procedures. AAEA was prepared by known procedures [61], while M1 and M2 were prepared and characterized as previously described [36]. Organic solutions were dried over anhydrous magnesium sulphate and were evaporated using a rotatory evaporator operating at a reduced pressure of about 10–20 mmHg. The melting ranges of solid compounds were determined on the instrument previously described [45]. ATR-FTIR analyses and potentiometric titrations were carried out using instruments and procedures previously reported [45]. ^1^H and ^13^C NMR, GC-MS, GC-FID, HPLC, and elemental analyses were carried out to confirm the structures of M1 and M2, as well as those of the intermediates synthesized during their preparations, obtaining results that were like those previously obtained and available in Alfei et al.’s recently published study [36]. The procedures and instruments used to perform such analyses, as well as the methods and materials employed to carry out column chromatography and thin layer chromatography during the preparation of M1 and M2, were previously described [36,45]. The optical microscopy analyses were performed using a Nikon Alphaphot-2YS2 microscope equipped with a hot stage cell (FP82HT Mettler, CH) and a 5 Mpixel live resolution digital microscopy camera (Moticam5 Motic, Canada). Image analysis and measurements were performed using Motic Images Plus 2.0ML software using a 4× objective. Sieving was performed with a 2000 Basic Analytical Sieve Shaker-Retsch apparatus (Retsch Italia, Verder Scientific S.r.l., Pedrengo (BG) Italy). Lyophilization and centrifugations were performed as previously described [43]. UV-Vis analyses were carried out using an Agilent Cary 100 UV/Vis Spectrophotometer (Agilent Technologies Italia S.p.A., Milan, Italy).

### 3.2. Synthesis of R1 and R2: General Procedure

A mixture of hexane and carbon tetrachloride (CCl_4_) was placed in a round-bottom cylindrical flanged reactor equipped with an anchor-type mechanical stirrer and nitrogen inlet, thermostated at 35 ± 0.05 °C and deoxygenated by nitrogen (N_2_) bubbling for 30 min. In the meantime, a solution was obtained by dissolving in a test-tube under N_2_ the monomer M1 or M2, DMAA, AAEA, and ammonium persulfate (APS) (1.8% wt/wt with respect to M1 + DMAA + AAEA and 2.6% wt/wt to M2 + DMAA + AAEA) in deoxygenated water distilled over KMnO_4_ and was siphoned into the reaction vessel. The density of the organic phase was adjusted by adding CCl_4_ so that the aqueous phase sank slowly when the stirring was stopped. The mechanical stirring was set to 900 rpm, SPAN 85 dissolved in hexane was added to the mixture, and the polymerization was started. After 10 min, N,N,N,N,-tetra-methyl-ethylene-diamine (TMEDA) was introduced, and the polymerization was continued. At the end of polymerization time, the resin was filtered, washed several times with 100 mL of a series of selected solvents (2-propanol, chloroform, water, absolute ethanol, chloroform, 2-propanol, and acetone in order), then dried at reduced pressure and r.t. for 16–20 h to achieve a constant weight. Table 8 contains data from two optimized polymerizations for preparing R1 and R2.

ATR-FTIR (R1) (ν, cm^−1^): 3399 (-NH_3_^+^), 2930 (CH_2_), 1609 (C=O).

ATR-FTIR (R2) (ν, cm^−1^): 3401 (-NH_3_^+^), 2927 (CH_2_), 1614 (C=O).

### 3.3. Sieving of R1 and R2

The dried resins (R1 and R2), after light grinding with a pestle to burst the fragile aggregates, were sieved by sieves with an external diameter (Ø_E_ = 10 cm) and 35–120 meshes and chosen after preliminary tests on small samples, using the analytical sieves described in Section 3.1. as a vibrating base.

### 3.4. Scanning Electron Microscopy (SEM)

The microstructure of resin R1 and R2 was also investigated by SEM analysis. In the performed experiments, the sample was fixed on aluminum pin stubs and sputter-coated with a gold layer of 30 mA for 1 min to improve the conductivity, and an accelerating voltage of 20 kV was used for the sample’s examination. The micrographs were recorded digitally using a DISS 5 digital image acquisition system (Point Electronic GmbH, Halle, Germany).

### 3.5. Estimation of -NH_3_^+^ Content in R1 and R2

The NH_2_ content of the resins R1_1 and R2_2 was estimated following the method of Gaur and Gupta [39]. Briefly, the two resins selected as representatives of the R1 or R2 families (ca. 2 mg) were taken in a 2 mL Pierce reaction vial, to which 4-O-(4,40-dimethoxytriphenylmethyl)-butyryl (0.25 mL) (reagent A), a catalytic amount of di-methylamino-pyridine (DMAP) (5 mg), and triethylamine (TEA) (100 µL) were added. The vial was screw-capped and gently tumbled at r.t. for 30 min. The reaction mixture was transferred to a 2-mL sintered funnel and washed successively with N, N-dimethylformamide (DMF) (2 × 10 mL), MeOH (2 × 10 mL), and, finally, with dry diethyl ether (2 × 10 mL). After the polymer support was dried under vacuum, a weighed quantity (ca. 1 mg) was taken in a 10 mL volumetric flask which was then filled up to the mark with HClO_4_ (reagent B). The released 4,40-dimethoxytriphenylmethyl cation (ε = 70,000) was estimated spectrophotometrically at 498 nm against reagent B as blank. From the values of absorbance (*A*) determined at 498 nm, the NH_2_ moles for grams of R1 and R2 were estimated using Equation (11).
(11)$$\frac{\mathrm{Moles}\;\mathrm{NH}2}{R1\;\mathrm{or}\;R2\left(g\right)}=\frac{A\times V}{p\times 70,000}$$

In the equation, *V* is the volume of the mixture used for detritylations (10 mL), and *p* is the weight in milligrams of the resin functionalized (1.5 mg) and subjected to detritylations.

### 3.6. Preparation of R1HG and R2HG: General Procedure

A certain volume of resins (Vi) was inserted in a graduated centrifuge tube (Ø_est_ = 14 mm, V = 10 mL), weighted (Wi), and added at r.t. and under magnetic stirring, with an excess of deionized water up to a total volume of 10 mL. When a homogeneous mixture was observed, the magnetic stirrer was removed, and the dispersions obtained were first gently shaken with a spatula for a few minutes to push away the trapped air, then sonicated at 37 °C for 30 min, and then degassed for 9 min using the Ultrasonic Cleaner 220 V and working at a frequency of 35 kHz with timer range of 1–99 min and temperature range of 20 to 69 °C (68 to 156 °F) (VWR, Milan, Italy). Upon centrifugation at 4000 rpm for 20 min, R1HG and R2HG, at their maximum content levels of water and EDS, were separated from water in excess, which was removed. The tube was then turned upside down on filter paper to remove residual water and so left for 10 min before measuring their final volumes (*Vf*), corresponding to the volume of gels at their EDS. Accordingly, the volume of water in which R1 and R2 resulted finally dispersed; once removed, the fraction of water that was not absorbed was determined. At this point, it was possible to calculate the concentrations (mg/mL) of R1 and R2 in the gels, which were expressed as mg/mL and percentages (%, wt/v) (Table 1). The initial volumes (*Vi*) and the final volumes (*Vf*) were used to determine the *EDS* percentages and the *EWC* percentages using Equations (12) and (13).
(12)$$EDS\;\left(\%\right)=\frac{Vf-Vi}{Vi}\times 100$$
(13)$$EWC\;\left(\%\right)=\frac{Vf-Vi}{Vf}\times 100$$

The gels were then left in their tubes, carefully sealed to prevent water evaporation, and stored in the fridge for subsequent characterization experiments, including ATR-FTIR, water loss, potentiometric titrations, optical analyses, and rheological studies.

### 3.7. ATR-FTIR Spectra

ATR-FTIR analyses were carried out both on R1 and R2, on the soaked gels (R1HG and R2HG), and on the gels obtained by gently heating R1HG and R2HG, while ATR-FTIR spectra were acquired on lyophilized samples of R1 and R2 after they were used to treat water solutions of F, MO, and MOF, and on F, MO, and MOF (nine spectra). The spectra were acquired from 4000 to 600 cm^−1^, with 1 cm^−1^ spectral resolution, co-adding 32 interferograms, with a measurement accuracy in the frequency data at each measured point of 0.01 cm^−1^ due to the internal laser reference of the instrument. Acquisitions were made in triplicate, and the spectra shown in Section 2.5 (Figure 4) and in the Appendix A are the most representative images. The spectral data obtained were then included in dataset matrices and were processed using the principal components analysis (PCA), by means of CAT statistical software (Chemometric Agile Tool, free down-loadable online, at: http://www.gruppochemiometria.it/index.php/software/19-download-the-r-based-chemometric-software; accessed on 27 November 2022). In particular, we arranged the data of the six ATR-FTIR spectra in a matrix of 3401 × 6 (n = 20,406) of measurable variables, while the ATR-FTIR data of the other nine spectra plus those of spectra of original R1 and R2 were arranged in a matrix of 3401 ×11 (n = 37,411). For each sample, the variables consisted of the values of transmittance (%) associated with the wavenumbers (3401) in the range of 4000–600 cm^−1^. The spectral data in the matrix were pretreated by SNV.

### 3.8. Weight Loss (Water Loss) Experiments

Exactly weighted samples of the swollen resins R1HG (683.0 mg) and R2HG (518.9 mg) were deposited in Petri dishes (PDs). The PDs were then placed in an oven under controlled temperature (37 °C), and the weight loss, meaning loss of water, was monitored as a function of time until a constant weight was reached. The cumulative weight loss percentages were determined by means of the Equation (14),
(14)$$\mathrm{Weight}\;\mathrm{loss}\;\left(\%\right)=\frac{MQ-Mt}{MQ}\times 100$$
where *MQ* and *Mt* are the initial mass of the swollen resin and its mass after time *t*, respectively.

### 3.9. Equilibrium Swelling Rate Experiments

The swelling measurements were carried out at r.t. by immersing exactly weighted samples of the dried resins R1 and R2 (231.8 mg and 164.2 mg, respectively), obtained from the polymerization reaction in deionized water (pH = 7.7 for R1 and 7.2 for R2) in a test tube. At fixed intervals of time [51], the samples in the test tube were centrifugated (15 min, 4000 rpm) to remove the not absorbed water, inverted on filter paper to absorb residual water, and weighed. The cumulative swelling ratio percentage (Q%) as function of time was calculated from Equation (15),
(15)$$Q\;\left(\%\right)=\frac{Ws-Wd}{Wd}\times 100$$
where *Wd* and *Ws* are the weights of the dried and swollen resin, respectively. The equilibrium swelling ratio (Q_equil._) was determined at the point when the hydrated resin achieved a constant weight.

### 3.10. Potentiometric Titration of R1HG and R2HG

Potentiometric titrations were performed on R1HG and R2HG at r.t., and the titration curves of gels were obtained. Exactly weighted samples of each gel (351.0 mg of R1HG and 337.4 mg of R2HG) were suspended in 50 mL of Milli-Q water (m-Q), then treated under magnetic stirring with a standard 0.1 N NaOH aqueous solution (2.0 mL, pH = 10.88 for R1HG and pH = 10.82 for R2HG). The solutions were potentiometrically titrated under stirring by adding aliquots (0.5 mL) of HCl 0.1N up to pH < 3 for a total volume of 5.0 mL [36,62]. Titrations were performed in triplicate, and measurements were reported as mean ± SD. The titration curves shown in the Section 2 are those obtained by plotting the data obtained by carrying out the representative experiment described here.

### 3.11. Rheological Studies

The rheological properties of R1HG and R2HG were assayed by a continuous shear method using a Brookfield viscometer (Viscostar-R, Fungilab S.A. Torino, Italy). In particular, samples of about 2 g were subjected to shear rates ranging from 1 to 100 s^−1^. All measurements were carried out at r.t. and were expressed as the mean values of five independent determinations. The image reported in the Section 2 is representative of one determination.

### 3.12. Adsorption of Dyes with R1 and R2

Methyl orange (MO) and fluorescein sodium salt (F) were chosen as the model compounds for anionic dyes, which can be found as water contaminants. The maximum wavelengths (𝜆 max) at 465 nm and 490 nm were used for MO and F, respectively, to determine the absorbencies using the UV-Vis spectrometer, as described in Section 3.1. The calibration curves of the dyes were achieved according to the procedure described in the following section.

#### 3.12.1. Calibration Curves

Stock solutions of dyes at concentrations of 500 mg/L were prepared by diluting 125 or 500 mg of the dye powders in 250 or 1000 mL of deionized water, respectively, in proper volumetric flasks. For MO, solutions at concentrations of 1, 2, 4, 5, and 10 mg/L were prepared by correct dilution from the stock solutions. Analogously, for F, solutions at concentrations 1, 2, 2.5, 4, and 5 mg/L were prepared. The absorbance for all prepared solutions was measured using a UV-Vis spectrophotometer at the proper 𝜆 max reported previously. The values of absorbance (abs) obtained were plotted against concentrations (mg/L), and the calibration curves reported in Appendix A were achieved, whose Equations (16) and (17) have been reported below.
(16)y=0.0764x+0.0100
(17)y=0.1894x+0.0176

Equations (13) and (14) were used for determining the concentrations of dyes in appositely prepared contaminated water samples before and after treatment with the resins reported here.

#### 3.12.2. Effect of Contact Time in Bulk Experiments

A total of 250 mL of MO and F solutions with dye concentrations of 50 mg/L were prepared in conical flasks. Additionally, a 250 mL of MO + F solution (MOF) was prepared by diluting to 250 mL, 25 mL of MO 500 mg/L, and 25 mL of F 500 mg/L. Aliquots of 5 mL were withdrawn before adding the absorbents, were opportunely diluted to adapt them to the calibration curves (MO and F 1:10, MOF 1:20), and the dye concentrations at time 0 (T0) were measured spectrophotometrically. Then, the residual 245 mL solutions were added with the adsorbents (R1 or R2) to reach R1 and R2 concentrations of about 1 g/L (250 mg). The suspensions were kept under magnetic stirring, and the MO and F concentrations in all suspensions were measured spectrophotometrically at the wavelengths corresponding to their maximum absorbance (λ max), after 5, 10, 20, 30, 60, 90, and 120 min. In particular, aliquots of 5 mL were withdrawn at the fixed time points, the dye solutions were separated from the adsorbent hydrogels by centrifugation, were diluted as the solutions at time T0, and the absorbance of solutions was then measured. The measurements were made in triplicate, and results of absorbance were reported as means ± SD.

The removal efficiency of R1HG and R2HG were calculated from Equation (18),
(18)$$R\%=\frac{Co-Ct}{Co}\times 100$$
where *Ct* and *Co* are the concentration of dye at any time and their initial concentrations (mg/L), respectively; and *R* is the removal percentage.

The amounts of dye adsorbed per unit mass of R1HG and R2HG (mg/g) at the fixed times (*Qt*) was determined according to Equation (19),
(19)$$Qt=\frac{\left[Co-Ct\right]\times V}{M}$$
where *Co* is the initial dye concentration (mg/L), *Ct* is the dye concentration (mg/L) at time *t*, *V* is the volume (L) of dye solution, and *M* is the mass of R1HG or R2HG (g). Finally, the amounts of adsorbed dye at equilibrium (*Qe*) by R1HG and R2HG were calculated from the mass balance Equation (20),
(20)$$Qe=\frac{\left[\left(Co-Ce\right)\times V\right]}{M}$$
where *Ce* and *Co* are the concentrations of dyes at equilibrium and their initial concentrations (mg/L), respectively, *V* is the volume of dye solution at equilibrium (L), and *M* is the mass of R1HG or R2HG (g).

#### 3.12.3. Filtration Adsorption Experiments

The filtration adsorption experiments were conducted using a plastic syringe as column (the volume capacity was 20 mL, and the diameter was 1.5 cm) loaded with R1HG or R2HG as filtrating material. Specifically, the hydrogels were prepared by dispersing R1 (0.8 mL, 595.2 mg) or R2 (0.7 mL, 498.3 mg) in an excess of deionized water directly in the syringe. The resins were allowed to absorb water under sonication (20 min), and once resins have reached their EDS, the excess of not absorbed water was removed first by gravity and then applying compressed air from the top. A final volume of 8 mL was reached. In the first experiment, separate solutions containing MO or F at concentrations 50 of mg/L were prepared and opportunely diluted (1:10), and the related UV-Vis absorbances were measured at their specific λ max. Then, as reported [54], a volume of original MO solution (MO concentration 50 mg/mL) equal to the volume of gels in the column (8 mL) was added from the top of the hydrogels and was filtered through the resins which were renewed before undertaking the filtration of the F solution (F concentration 50 mg/L). Compressed air from the top of the column was applied to promote the complete emptying of the column. The filtrates were collected separately from the bottom of the column, diluted as described above (1:10), and analyzed by UV-Vis. In a subsequent experiment, the solutions of MO and F (50 mg/L) were successively filtered on the same resin without refreshing it. The filtrates were collected separately, diluted as described above (1:10), and analyzed by UV-Vis. Finally, 8 mL of a single solution containing a mixture of MO and F was prepared by diluting 10 mL 1 mL of MO 500 mg/L, and 1 mL F 500 mg/L was filtered through R1HG and R2HG. The concentrations of dyes in this solution were measured spectrophotometrically after proper dilution (1:20). Then, the solution was added to the column and filtered, and the filtrate was diluted (1:20) and analyzed, as in the previous experiments. The measurements were made in triplicate, and results of absorbance were reported as means ± SD. The removal efficiency of R1HG and R2HG in all of the experiments was calculated from Equation (21),
(21)$$R\%=\frac{Co-Cf}{Co}\times 100$$
where *Cf* and *Co* are the concentrations of dyes in the filtrate and the initial concentrations (mg/L), respectively.

## 4. Conclusions

In this work, we wanted to contribute to the search for promising, low-cost, and efficient solutions for the treatment of industrial wastewater contaminated by organic anionic dyes, which is one of the most widespread problems afflicting people throughout the world.

Without recovering to complex biodegradation processes, that occur with the formation of metabolites, often more toxic than the dye to remove, we thought to develop materials to be used in adsorption processes, which could be able to absorb and retain anionic dyes by means of electrostatic interactions, as also confirmed by kinetic studies.

To this end, monomers M1 and M2, which were previously reported, were synthesized again and copolymerized with DMAA via a low-cost, operator-friendly, one-step reverse-phase suspension copolymerization technique using AAEA as a cross-linker and achieving cationic resins R1 and R2. Once characterized by several analytical techniques, upon their dispersion in an excess of water, R1HG and R2HG hydrogels were achieved without using any other additive or gelling agent, which could be released and paradoxically contaminate water during the use of R1HG or R2HG as decontaminant adsorbents. R1HG and R2HG demonstrated high EDS and EWC, a pseudoplastic/Bingham plastic shear thinning behavior, and they release water following first-order kinetics (R1) and Korsmeyer–Peppas kinetics (R2) when gently heated. Both R1 and R2 and the related hydrogels demonstrated a high content of -NH_3_^+^ groups for grams, which is essential for an efficient absorbent activity by electrostatic interactions. In fact, in absorption experiments carried out in different conditions and monitored with UV-Vis methods, R1HG and R2HG showed high adsorption efficiency based on electrostatic interactions, both by contact (97–100%) and by filtration (100%) towards the azo dye methyl orange (MO) and fluorescein sodium salt (F), which were selected as models of anionic dyes, and their mixture (MOF). The results from kinetic studies established that the kinetic behavior of the absorption of F, MO, and of F and MO in MOF mixture using both R1 and R2 as adsorbents followed the pseudo-second-order kinetic model. In this regard, the values of K2 established that R2 was able to remove F, MO, and F + MO faster than R1 and that the removal of F was more rapid than that of MO, both when F and MO were treated alone and when they were in a mixture. Collectively, R1 and R2 and their gels, R1HG and R2HG, represent low-cost and up-scalable materials that are promising for applications in industrial wastewater treatment.

## Data Availability

All data concerning this work are included in the present manuscript and in the Appendix A associated with this paper.

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
