# Peer review of "Cationic Polystyrene-Based Hydrogels as Efficient Adsorbents to Remove Methyl Orange and Fluorescein Dye Pollutants from Industrial Wastewater"

_ijms, 2023, doi:10.3390/ijms24032948_

Round 1
Reviewer 1 Report
In the present study, cationic hydrogel based on polystyrene was synthesized and used as an efficient adsorbent to remove dyes from aqueous solution. The study has good results and it is suggested to be accepted after making the following corrections:
- In the abstract section, the experimental results, the maximum adsorption capacity and the optimal value of the effective parameters in the surface adsorption process should also be presented.
- It is necessary to provide the type of color used in the title of the mentioned article.
- In the introduction section, information about the importance and reasons for using hydrogel in the process of cleaning aqueous solutions should also be presented.
- In the introduction section, the effects of colors in the environment, various methods of their removal should also be presented.
- Figures 2 and 3 are not suitable and should be removed and SEM images should be used instead.
- FTIR analysis should be checked after the surface adsorption process of the desired colors.
- Appropriate reference should be provided for the mentioned functional groups in the structure of adsorbents.
- Figure 12 is not necessary and it is suggested to remove it and use more important images instead.
- It is suggested that the conditions for effective parameters in the adsorption process are presented below the relevant diagrams.
- It is suggested to study the kinetics of the adsorption process.
- It is necessary to perform BET analysis and determine the specific active surface for adsorbents.
- It is suggested that the introduction section be improved using the following articles.
- doi.org/10.1016/j.matchemphys.2022.126088, doi.org/10.1016/j.envres.2022.113349, doi.org/10.3390/nano12183103, doi.org/10.1016/j.matchemphys.2021.125655, doi.org/10.1007/s10924-022-02623-x
- The quality of the English presentation of the article should be improved.
Author Response
Reviewer 1
In the present study, cationic hydrogel based on polystyrene was synthesized and used as an efficient adsorbent to remove dyes from aqueous solution. The study has good results and it is suggested to be accepted after making the following corrections:
- In the abstract section, the experimental results, the maximum adsorption capacity and the optimal value of the effective parameters in the surface adsorption process should also be presented.
We thank the Reviewer for his suggestion, with which we agree. The required data have been included in the Abstract (lines 28-34).
- It is necessary to provide the type of color used in the title of the mentioned article.
The type of dyes used in our study has been included in the title of the manuscript, thus addressing the request of the Reviewer (line 3).
- In the introduction section, information about the importance and reasons for using hydrogel in the process of cleaning aqueous solutions should also be presented.
The requested information and the related references (Refs. 31-34) have been included in the Introduction section as suggested (lines 135-146).
- In the introduction section, the effects of colors in the environment, various methods of their removal should also be presented.
Although some dangerous effects of dyes in the environment were already present in the original paper, other ones with related reference (Ref. 8) have been included in the revised manuscript, to satisfy the Reviewer request (lines 52-65). Concerning various methods used for the removal of dyes, they were already present in the not revised manuscript. Please, see at lines 76-88, 94-103 and 117-132. However, to meet the request of the Reviewer, additional methods with the related references have been inserted in the revised manuscript (lines 85-93, 104-109, 113-116 and 147-149).
- Figures 2 and 3 are not suitable and should be removed and SEM images should be used instead.
The request of the Reviewer has been satisfied. SEM images of resin R2 have been acquired and SEM images of both R1 and R2 have been included in the work. Particularly, the SEM image of R1 has been included in the Supplementary Materials, since already published in our recent work (Ref. 38, revised manuscript), while two representative SEM images of R2 have been included in main text associated to Figure 2, as Figure 2c and 2d. We kindly ask the Reviewer to not force use to remove the optical images in Figure 2, which we would keep. Concerning Figure 3, it should be maintained because concerns the optical images of the swollen resins (hydrogels) on which SEM cannot be carried out.
- FTIR analysis should be checked after the surface adsorption process of the desired colors.
ATR-FTIR analyses of the resins after the surface adsorption process were acquired on R1/F, R1/MO, R1/MOF and on R2/F, R2/MO, R2/MOF. Additionally, for comparison we acquired also the ATR-FTIR spectra of F, MO and MOF. Images of the acquired spectra have been included in the Supplementary Materials as Figure S15 and discussed in the main text (lines 622-636). Moreover, the spectral data of the new acquired spectra plus those of R1 and R2, were arranged in a matrix and processed by PCA. PCA results were reported as score plot of PC1 vs. PC2 in the new Figure S16 in Supplementary Materials, and were discussed in the main text (lines 642-652)
- Appropriate reference should be provided for the mentioned functional groups in the structure of adsorbents.
The requested reference has been included (Ref 11) in the Introduction section (line 73).
- Figure 12 is not necessary and it is suggested to remove it and use more important images instead.
As asked Figure 12 has been removed from the main text and included in the Supplementary Materials, as Figure S9.
- It is suggested that the conditions for effective parameters in the adsorption process are presented below the relevant diagrams.
As suggested, the conditions for effective parameters in the adsorption process have been included in the captions of Figure 12, 14 and S14 (revised material).
- It is suggested to study the kinetics of the adsorption process.
As suggested, the kinetics of the absorption process have been studied fitting to the absorption curves the pseudo-first order kinetic model, pseudo-second order kinetic model and intra-particle diffusion kinetic model. The discussion of results has been added (lines 653-692).
- It is necessary to perform BET analysis and determine the specific active surface for adsorbents.
We agree with the Reviewer that for hydrogels that remove dyes by absorption, as those reported in his relevant work we have cited (doi.org/10.1007/s10924-022-02623-x) is essential to determine the specific surface area, the pore volume and the pore size by BET analysis, but as reported in the main text, our cationic hydrogels remove anionic dyes by electrostatic interaction between the cationic ammonium groups and anionic sulphate or carboxylate groups of dyes. Consequently, to know the active groups in the hydrogels becomes essential to determine the content of NH3+ groups for gram of hydrogel. In this regard, in our work, such content has been determined both on dried resins and on swollen gels.
- It is suggested that the introduction section be improved using the following articles.
- doi.org/10.1016/j.matchemphys.2022.126088, doi.org/10.1016/j.envres.2022.113349, doi.org/10.3390/nano12183103, doi.org/10.1016/j.matchemphys.2021.125655, doi.org/10.1007/s10924-022-02623-x
All the papers suggested by the Reviewer have been cited in the Introduction section.
- The quality of the English presentation of the article should be improved.
We thank the Reviewer for his suggestion. The manuscript has been revised by Professor Deirdre Kants, English mother tongue working for the University of Genoa and Pavia, who helped us to detect and remove typos and English grammar errors.
Reviewer 2 Report
The present manuscript reports on the “Cationic Polystyrene-Based Hydrogels as Efficient Adsorbents to Remove Dyes Pollutants from Industrial Wastewater”. The work is of some interest but seems to be too primitive and lacks novelty, proper scientific support and justification. The synthesized microgel is not sufficiently characterized to support the claims. There are many reports exhibiting adsorption process. Thus, in my opinion, the manuscript in its present form cannot be considered for publication. I recommend major revision.
Following are some of the comments/suggestions which will be useful to the authors.
1. First of all, there are many previous works published for adsorption process. The authors seem deliberately avoid those papers. This is unusual, as the authors need to acknowledge the previous literature and compare their work with the similar ones in the literature and demonstrate their research outcomes in terms of advantages and disadvantages. Some of studies are given below need to cited;
1) Arif, M. (2022). Complete life of cobalt nanoparticles loaded into cross-linked organic polymers: a review. RSC Advances, 12(24), 15447-15460.
2) Arif, M., Shahid, M., Irfan, A., Nisar, J., Wang, X., Batool, N., ... & Begum, R. (2022). Extraction of copper ions from aqueous medium by microgel particles for in-situ fabrication of copper nanoparticles to degrade toxic dyes. Zeitschrift für Physikalische Chemie, 236(9), 1219-1241.
3) Arif, M. (2023). Extraction of iron (III) ions by core-shell microgel for in situ formation of iron nanoparticles to reduce harmful pollutants from water. Journal of Environmental Chemical Engineering, 109270.
2. NH3+ is not correct as written in lines 22, 172 and other places. Make a bond or convert it in NH4+.
3. Why peak intensity of R2HG is very weak than R2 and D-R2HG in Figure 4.
4. Write complete name of synthesized microgel with abbreviation. Correct it from line 14.
5. It is not possible that you use crosslinker and a simple polymer product is formed. There is maximum possibility to form microgels. What is in your case?
6. What is MOF? Is metal organic framework? It is very confusing abbreviation because MOF stand for metal organic framework.
7. Write the adsorption conditions in the Figure 6 to 14. The conditions such as pH, mixing time, content of adsorbent and adsorbate are directly affected on adsorption rate. Therefore, it is very important to write these conditions.
8. Maximum percentage removal occurred immediately after adding the adsorbent but after that the percentage removal decreased as given in Figure 13. Why?
9. The explanation of results is not sufficient. Mostly the results are given but discussing portion is missing.
10. Improve the conclusion portion of this manuscript.
11. Improve the title of the manuscript.
12. Where are the results of 1H and 13C NMR, GC-MS, GC-FID, HPLC, 614 elemental analyses? Also write the complete model and company names of instruments.
Author Response
Reviewer 2
The present manuscript reports on the “Cationic Polystyrene-Based Hydrogels as Efficient Adsorbents to Remove Dyes Pollutants from Industrial Wastewater”. The work is of some interest but seems to be too primitive and lacks novelty, proper scientific support and justification. The synthesized microgel is not sufficiently characterized to support the claims. There are many reports exhibiting adsorption process. Thus, in my opinion, the manuscript in its present form cannot be considered for publication. I recommend major revision.
Following are some of the comments/suggestions which will be useful to the authors.
- First of all, there are many previous works published for adsorption process. The authors seem deliberately avoid those papers. This is unusual, as the authors need to acknowledge the previous literature and compare their work with the similar ones in the literature and demonstrate their research outcomes in terms of advantages and disadvantages. Some of studies are given below need to cited;
1) Arif, M. (2022). Complete life of cobalt nanoparticles loaded into cross-linked organic polymers: a review. RSC Advances, 12(24), 15447-15460.
2) Arif, M., Shahid, M., Irfan, A., Nisar, J., Wang, X., Batool, N., ... & Begum, R. (2022). Extraction of copper ions from aqueous medium by microgel particles for in-situ fabrication of copper nanoparticles to degrade toxic dyes. Zeitschrift für Physikalische Chemie, 236(9), 1219-1241.
3) Arif, M. (2023). Extraction of iron (III) ions by core-shell microgel for in situ formation of iron nanoparticles to reduce harmful pollutants from water. Journal of Environmental Chemical Engineering, 109270.
We apologize with the Reviewer for not having considered his relevant works, here suggested. Now all works suggested have been cited in our manuscript.
- NH3+ is not correct as written in lines 22, 172 and other places. Make a bond or convert it in NH4+.
The Reviewer is right. So, we checked carefully all the manuscript finding ten points we needed correction. As suggested by the Reviewer a bond was inserted (lines 23, 171, 224, 225, 586, 590, 791, 792, 805, and 997).
- Why peak intensity of R2HG is very weak than R2 and D-R2HG in Figure 4.
We thank the Reviewer very much for his attention and comment. The bands concerning organic substances are very small in the IR spectrum of R2HG compared to those in the spectra of R2 and D-R2HG because (as widely explained in the text), R2HG is the hydrogel derived from R2 at its maximum level of swelling due to water absorption. Consequently, in the IR spectrum of R2HG, the water bands predominate over those of the organic material. In contrast, the dry resin obtained by reverse suspension polymerization R2 and the fully dried D-R2HG do not contain water. Consequently, the spectra of these substances that do not contain water show only the bands of their organic functional groups which are intense.
- Write complete name of synthesized microgel with abbreviation. Correct it from line 14.
As asked the complete names of hydrogels was inserted at the first mention of their abbreviations as also described in the instruction for authors of IJMS (line 19).
- It is not possible that you use crosslinker and a simple polymer product is formed. There is maximum possibility to form microgels. What is in your case?
We thank the Reviewer for his comment which gave us the possibility to better explain our case. As already described in the original manuscript, our case consisted of using crosslinked copolymers (R1 and R2), dispersing them in excess of water, leaving resins to absorb the maximum quantity of water under sonication, and finally removing the not absorbed water. Particularly, as reported in the experimental section and related discussion, we first copolymerized monomers M1 and M2 with DMAA in the presence of the crosslinker AAEA achieving the solid resins R1 and R2, insoluble in water, thus confirming that the crosslinking was successful. Surely, a lot of different products, with chains of different length, different molecular weight, and with particles of different size were present in the crude products. In fact, as described in the experimental section and in the related discussion, the crude products were subjected to sieving to reduce the distribution of particle size achieving R1 and R2 which were completely characterized by different techniques. Then, microgels were obtained by dispersing solid R1 and R2 in excess of water without adding other additives.
- What is MOF? Is metal organic framework? It is very confusing abbreviation because MOF stand for metal organic framework.
MOF stands for Methyl Orang + Fluorescein. Anyway, we agree with the reviewer that a specification was necessary. The due specification has been added at the first mention of MOF (lines 27-28.
- Write the adsorption conditions in the Figure 6 to 14. The conditions such as pH, mixing time, content of adsorbent and adsorbate are directly affected on adsorption rate. Therefore, it is very important to write these conditions.
As asked, the required information has been inserted in the captions of the suggested Figures, where necessary.
- Maximum percentage removal occurred immediately after adding the adsorbent but after that the percentage removal decreased as given in Figure 13. Why?
We thank the Reviewer for his answer. The phenomenon can be promptly explained.
Such fluctuation could be due to the UV-Vis instrument which, in fact, provides absorbance values with standard deviations, which also justify the presence of values over 100. Anyway, performing a t Student test was established that there was no significant difference between percentages obtained after reaching the maximum values and the maximum value itself. Such explanation has been added in the text (lines 598-604).
- The explanation of results is not sufficient. Mostly the results are given but discussing portion is missing.
More explanations have been included in the main text. Please, see the several changes present in Section 2.
- Improve the conclusion portion of this manuscript.
As asked, conclusions have been improved. Please, reconsider the revised version of Section 4.
- Improve the title of the manuscript.
As asked, the title has been modified and improved.
- Where are the results of 1H and 13C NMR, GC-MS, GC-FID, HPLC, 614 elemental analyses? Also write the complete model and company names of instruments.
The reviewer is right in his comment because we were poorly clear. The analysis he mentioned were not performed on the compounds reported in this manuscript, but on the intermediates synthetized previously (Ref. 37) to prepare M1 and M2 and on M1 and M2 themselves (Ref. 37). In the present work, we re-synthetized M1 and M2 and used the same instruments to characterize intermediates and final compounds obtaining results which confirmed their structure. Anyway, we retained redundant inserting such results here and we only cited the article where readers can find all. Similarly, the complete model and company names of such instruments was avoided to avoid duplications as suggested by the Editorial office and we only cited the articles where such information has been provided. Additional explanation regarding this question have been included in the text (lines 752-763).
Round 2
Reviewer 2 Report
Accepted